# Conservation of carbon resources and values on public lands: A case study from the National Wildlife Refuge System

**Zhiliang Zhu**[1]*, **Beth Middleton**[2], **Emily Pindilli**[3], **Darren Johnson**[2], **Kurt Johnson**[4], **Scott Covington**[4]

1 U.S. Geological Survey, Reston, VA, United States of America, 2 U.S. Geological Survey, Wetland and Aquatic Research Center, Lafayette, LA, United States of America, 3 U.S. Geological Survey, Science and Decisions Center, USGS Northeast Region, Reston, VA, United States of America, 4 U.S. Fish and Wildlife Service, Falls Church, Virginia, United States of America

* zzhu@usgs.gov

**Data Availability Statement:** This study used existing data that have been previously published, see link below. This study does not produce new data other than presented in tables and in the

## Abstract

Public lands in the United States are those land areas managed by federal, state, and county governments for public purposes such as preservation and recreation. Protecting carbon resources and increasing carbon sequestration capacity are compatible with public land management objectives for healthy and resilient habitats, i.e., managing habitats for the benefit of wildlife and ecosystem services can simultaneously capture and store carbon. To evaluate the effect of public land management on carbon storage and review carbon management as part of the land management objectives, we used existing data of carbon stock and net ecosystem carbon balance in a study of the National Wildlife Refuge System (NWRS), a public land management program of the U.S. Fish and Wildlife Service (Service). Total carbon storage of the 364 refuges studied was 16.6 PgC, with a mean value 42,981 gCm$^{-2}$. We used mixed modeling with Bonferroni adjustment techniques to analyze the effect of time since refuge designation on carbon storage. In general, older refuges store more carbon per unit area than younger refuges. In addition to the age factor, carbon resources are variable by regions and habitat types protected in the refuges. Mean carbon stock and the rate of sequestration are higher within refuges than outside refuges, but the statistical comparison of 364 refuges analyzed in this study was not significant. We also used the social cost of carbon to analyze the annual benefits of sequestrating carbon in these publicly managed lands in the United States, which is over $976 million per year in avoided $CO_2$ emissions via specific conservation management actions. We examine case studies of management, particularly with respect to Service cooperation activities with The Conservation Fund (TCF) *Go Zero*® Program, Trust for Public Land (TPL) and individuals. Additional opportunities exist in improving techniques to maximize carbon resources in refuges, while continuing to meet the core purpose and need of the NWRS.

supplemental material. https://www.sciencebase.gov/catalog/item/5e860f5ce4b01d50927fb724.

**Funding:** The author(s) received no specific funding for this work.

**Competing interests:** NO authors have competing interests

# 1. Introduction

## 1.1. Management and carbon in public lands

Publicly managed land and water represent 28% of the total area of the United States [1], and present an opportunity for protecting and enhancing carbon resources [2, 3] in addition to other benefits (e.g., recreation, hunting, fishing, storm mitigation, and water filtration) [4]. Public lands in the conterminous U.S. sequester an average of 53.2 TgC per year in biomass and soils, offsetting approximately 15% of fossil fuel emissions produced by the U.S. [5]. Protecting carbon resources and carbon sequestration capacity is compatible with core public land management objectives, particularly if carbon management is considered in long-term conservation strategies [6, 7].

National Wildlife Refuges (NWRS, Fig 1) are managed by the U.S. Fish and Wildlife Service to protect habitat following laws such as the National Wildlife Refuge System Improvement Act, Migratory Bird Treaty Act, and the Endangered Species Act. At the same time, carbon management provides ancillary benefits to conservation actions already being deployed in

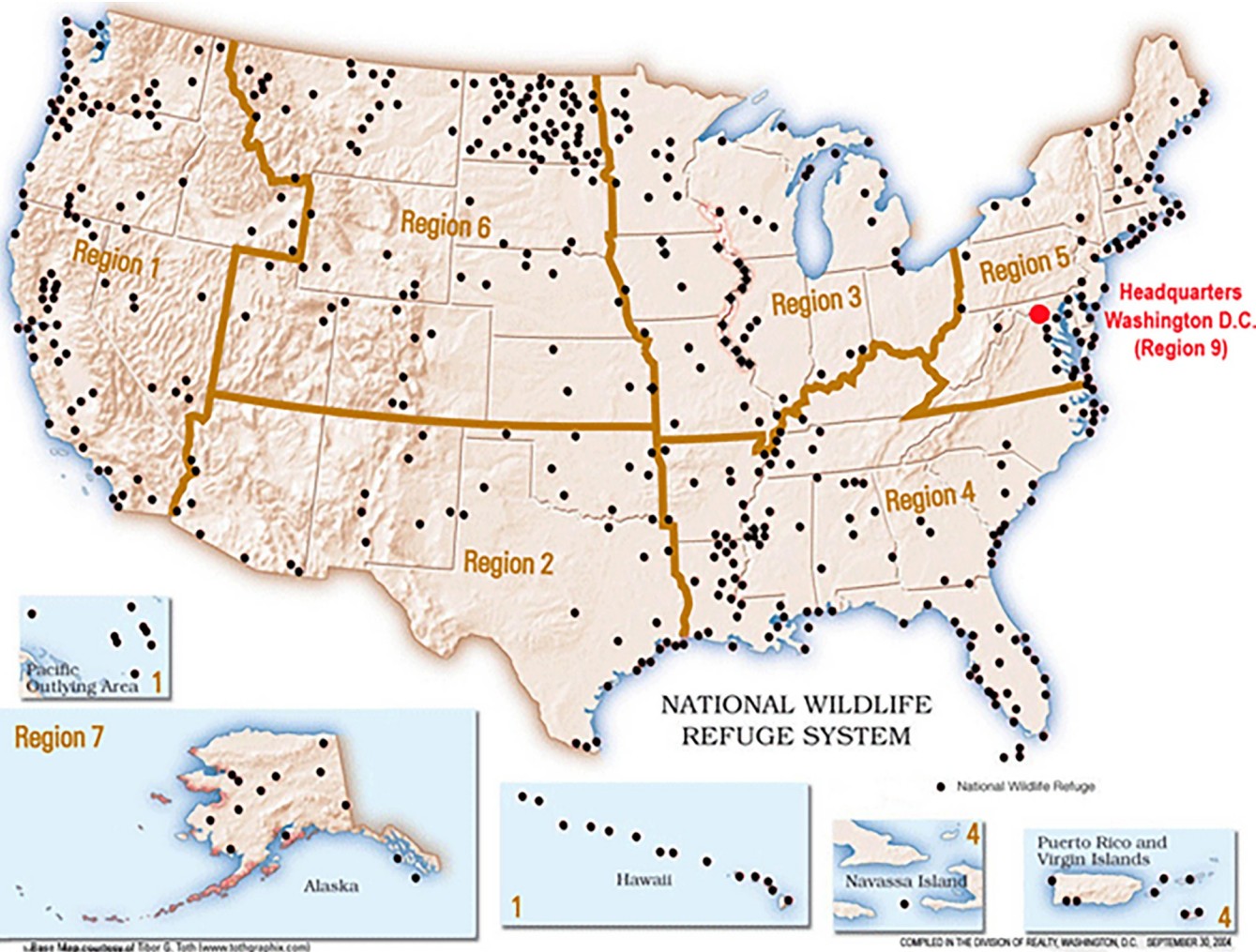

**Fig 1. U.S. Fish and Wildlife Service regional map showing locations of National Wildlife Refuge in 2019 (location and regional boundary data from Vandegraff 2005).**

these refuges. Federal protection and conservation of such habitats reduce land-use change and disturbances, which further enhance the protection of carbon resources [8]. For example, as part of a collaborative U.S. Geological Survey (USGS)-Service study conducted at the Great Dismal Swamp NWR [9], determined that wetlands restored for desired peatland conditions to benefit wildlife also sequester 200,000 tons of carbon per year, offsetting the annual emissions of 42,000 vehicles.

## 1.2 Societal benefits of carbon resources

The societal benefits associated with carbon resources arise from avoided carbon dioxide ($CO_2$) emissions and removal of carbon from the atmosphere (sequestration) also reduce the effects of climate change. Therefore, the conservation of public lands could result in a reduction of atmospheric $CO_2$ with associated benefits for society [2]. The economic value of avoided $CO_2$ emissions can be expressed as the social cost of carbon (SCC), which is an estimate of the net present value of avoided economic and societal damages associated with a one-ton increase (or decrease) in $CO_2$ emissions in a given year [10]. The value is based on projected damages including "changes in net agricultural productivity, human health, and property damages from increased flood risk and changes in energy system costs, such as reduced costs for heating and increased costs for air conditioning [10]. Damages associated with future climate change have been valued using various approaches with literature values ranging from $1 to $100 per ton of $CO_2$ [11, 12].

## 1.3. Management enhancement of wildlife habitat

Spatial and temporal change in wildfire, drought and hydrologic regimes, as well as land use, are major management concerns for public land managers [4], affecting ecosystem services and overall resilience of these ecosystems. These same factors also affect the storage and cycling of carbon resources in various carbon pools. For example, a single catastrophic wildfire in the Great Dismal Swamp NWR (Fig 2) reduced the mean elevation of the peat surface by 47 cm and released 1.10 Tg of carbon from a 25 km$^2$ burned area [13]. Pocosin Lakes NWR experienced a severe wildfire in 1985 consuming 40 km$^2$ and 7.0 TgC [14]; years later fires returned in 2008 and 2011 [15]. These events suggest that catastrophic fires are likely more common and more intense because of peat drying following the drainage of these wetlands, perhaps exacerbated by a changing climate.

We propose that management that promotes healthy and resilient habitats also protects carbon resources and sequestration capacities by limiting disturbances. Managing for ecosystem health (i.e., protecting and restoring forests, wetlands, and coastal systems) allows NWRS to contribute to climate mitigation by enhancing plant production and soil carbon storage [6, 7]. Conceptually, the soil's sequestration capacity would increase after refuge designation as native vegetation develops on formerly disturbed areas. The climate mitigation potential of carbon storage has also provided a rationale for investments in restoration of bottomland hardwood forest [16], eastern peatlands [17], blue carbon habitats (i.e., marine and coastal tidal wetlands) [18] and "teal carbon" of inland wetland systems [19, 20]. Finally, many refuge wetlands have engineered structures used to manipulate hydrology to benefit wildlife habitat while potentially improving carbon conservation.

In this paper, we draw from existing data to determine the amounts of carbon resources among various habitat types protected in refuges since establishment and highlight the economic value and climate benefit of the refuges. We consider the role of carbon conservation as one of the many benefits provided by public lands, in addition to wildlife habitat, improving water quality, and providing recreation and public access. Using several USGS-Service

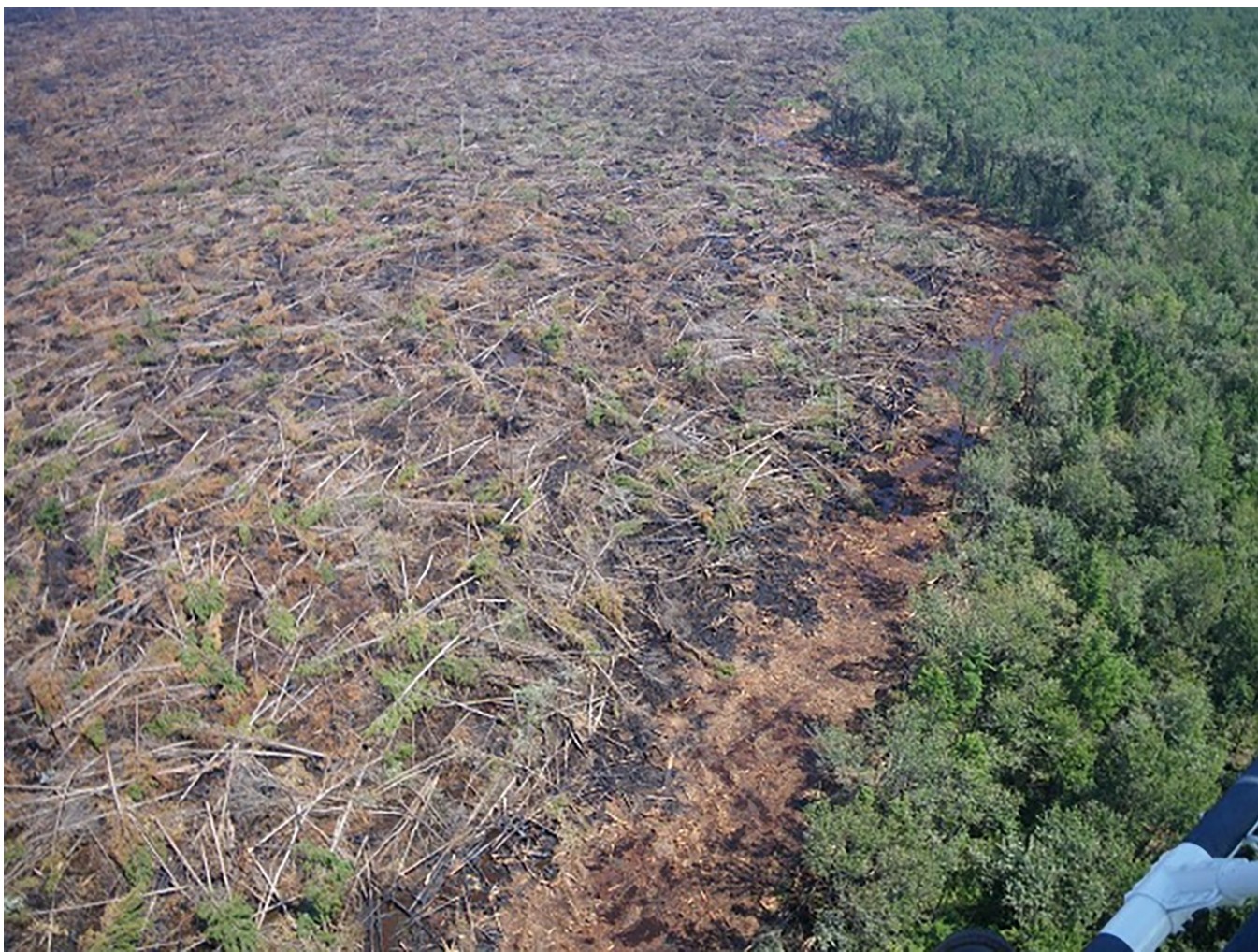

**Fig 2. An aerial photo of the aftermath of 2011 lateral fire that burned 2,700 ha of peatland swamp ecosystem at the great dismal swamp NWR.** Photo credit: U.S. Fish and Wildlife Service.

collaborative studies as examples, we illustrate the carbon sequestration opportunities associated with various ecosystems (e.g., terrestrial forests, fresh and saltwater wetlands, and Alaska permafrost). This information may be useful as part of management strategies to increase carbon capture and sequestration with current knowledge of carbon storage and accrual over time. Specifically, public land managers and stakeholders may be interested in our objectives designed to explore questions such as:

1. What are the amounts of carbon resources currently stored in various NWRS wildlife habitats, and how do these resources change over time under public land protection status?

2. What are the economic values of these resources to the public?

3. What are the opportunities and challenges under current NWRS land management practices for increasing carbon resources, avoiding losses, and accruing benefits to the public over time? How might these opportunities and challenges be built into existing management plans to support decision making at the scale where decisions are made on the ground?

Public lands have traditionally been managed to support a variety of uses including conservation and recreation. Increasingly they are also being looked upon to provide benefits through climate change mitigation via carbon sequestration and reduction of greenhouse gas emissions. The purpose of this study was to add useful information to this new management objective.

## 2. Materials and methods

### 2.1. Carbon resources data for refuges

To obtain estimates of both carbon stock (combining above- and below-ground carbon pools) and net ecosystem carbon balance (NECB) [21], by dominant habitat for each NWR, existing data products were used including map products of carbon stock and flux, refuge boundaries and land cover. Note that NECB is the net rate of carbon accumulation or loss over a specific time interval and is specific to an ecosystem (habitat) type. To determine the area of potential carbon resources in refuges, we used an online carbon data list, as described in a series of reports [22], with currently managed NWRS. Of the 592 refuges from the online refuge boundary data [23], 364 were included in our final dataset, each serving as one data point. Excluded refuges included locations of: 1) possible future acquisitions, 2) that fell outside the extent of the online carbon data list (e.g., islands or coastlines), 3) with area size was smaller than 4 km$^2$, noting that the raster resolution of the carbon data was 4 km$^2$. NWRS were historically grouped into eight regions (location and regional boundary data from [24]; Table 1). For our purposes, we considered Hawaii separately because it was ecologically and geographically distinct from the contiguous United States (CONUS).

Wildlife habitat types were derived from the 2011 National Land Cover Database (NLCD [25]), which were developed based on Landsat imagery with 30-meter resolution. The 20 NLCD map classes [25] were grouped into four dominant habitat types: forest (all three NLCD forest classes and woody wetland class), wetland (herbaceous only), grassland (all grass and shrub cover classes), and other (the rest of the NLCD classes). While the groupings were consistent with the use of NLCD data in the U.S. national greenhouse gas inventory mapping convention [26], the grouping of detailed land cover classes into few habitat types represented a practical approximation. The resulting four map classes were intersected with NWR boundary database to obtain a shapefile of land cover types and sizes for each NWR in their respective

**Table 1. Total area (km$^2$), mean carbon stock (gCm$^{-2}$), and mean net ecosystem carbon balance (NECB, gCm$^{-2}$yr$^{-1}$) of National Wildlife Refuges considered in this study by FWS regions and dominant habitat types.**

| Region | Forest | | | Grassland | | | Wetland | | | Other | | | Regional summary | | |
|---|---|---|---|---|---|---|---|---|---|---|---|---|---|---|---|
| | Area | Stock | NECB | Area | Stock | NECB | Area | Stock | NECB | Area | Stock | NECB | Area | Stock | NECB |
| Pacific | 237 | 14,941 | 241 | 4,450 | 2,946 | 16 | 1,123 | 4,738 | 39 | 175 | 9,636 | 121 | 5,986 | 3,954 | 33 |
| Southwest | 857 | 7,999 | 176 | 8,897 | 1,416 | 19 | 1,541 | 5,622 | 99 | 684 | 3,195 | 47 | 11,978 | 2,529 | 42 |
| Midwest | 1,623 | 12,527 | 95 | | | | 634 | 15,454 | -16 | 1,294 | 8,670 | 51 | 3,551 | 11,644 | 59 |
| Southeast | 8,514 | 17,293 | 245 | | | | 2,184 | 15,379 | 37 | 1,519 | 9,161 | 126 | 12,217 | 15,940 | 193 |
| Northeast | 1,529 | 20,980 | 138 | | | | 363 | 15,689 | 85 | 144 | 10,355 | 140 | 2,035 | 19,287 | 129 |
| Mountain Prairie | 37 | 13,549 | -15 | 6590 | 3,329 | -9 | 60 | 4,987 | 33 | 1,280 | 4,730 | -14 | 7,968 | 3,614 | -9 |
| Alaska | 87,814 | 23,347 | -7 | 244,395 | 57,319 | 9 | | | | 1,646 | 109,889 | 38 | 333,854 | 48,643 | 5 |
| Pacific Southwest | 17 | 10,647 | 270 | 7,476 | 1,648 | 34 | 918 | 5,653 | 98 | 476 | 9,713 | 192 | 8,887 | 2,511 | 50 |
| Hawaii | 198 | 49 | 0 | 4 | 41 | 259 | | | | 4 | 39 | 344 | 206 | 49 | 12 |
| Habitat summary | 100,827 | 22,424 | 20 | 271,813 | 51,758 | 10 | 5,432 | 11,844 | 68 | 8,611 | 26,272 | 57 | 386,682 | 42,981 | 14 |

A negative value of NECB designates carbon source. Blank table cell indicates that the habitat type does not exist in the region from the data used.

FWS regions (Table 1). In addition to the 364 NWR habitat data points, we also obtained the same habitat data of similar sized polygons that were adjacent to each of the refuges for the purpose of examining any differences inside vs. outside of the refuge.

The two carbon variables (stock and NECB) were spatially overlain with the habitat types within the actual boundary of each NWR. We obtained the year of refuge' designation from FWS online source [27] and calculated how many years each of the refuges had been established as of 2020. Out of the 364 refuges, the oldest refuge was designated 116 years ago (Breton NWR, Southeast Region) and the youngest was designated 8 years ago (Wapato Lake, Pacific Region). The average age of refuge was 57 years. The two carbon variables by the 364 refuges by FWS regions were summarized in Table 1 and S1 Table. Zonal statistics was used to determine the mean (that is, value per square meter annually) estimates of carbon flux and stock for each NWR by the habitat types.

## 2.2. Statistical analysis

Effects of refuge designation on carbon stock and NECB were analyzed by FWS regions, habitat types, refuge sizes (area size of refuge), latitude and longitude of the center of the refuge, and years since the refuge designation or establishment. The same analysis was conducted for habitats within refuge boundaries as well as lands immediately outside of the refuges in order to test for effects of the conservation management. Carbon estimates and year since designation were log-transformed in order to analyze age effect on carbon accumulation, which has a high range of values, and compared across dominant habitat types nested within the FWS regions to meet normality assumptions of analysis of variance (ANOVA). After transformation, the residuals met the assumptions of homogeneity, all final models met the assumption of linearity.

The models tested included a) area of refuge as a function of total area by region, latitude, longitude and the nested ANOVA model; b) mean carbon stock as a function of total area by region and the nested ANOVA model; c) mean of NECB as a function of the nested model.

The nested ANOVA model consisted of region, dominant habitat within region, inside and outside of the refuge, dominant habitat within region as well as the interactions between inside and outside with region and dominant habitat within region.

We used mixed modeling with Bonferroni adjustment techniques in SAS software (PROC Mixed; SAS V 9.3; [28]) to make comparisons of multiple effects [29]. Least square means of significant interactions were compared following Bonferroni adjustments of significant differences.

## 2.3. Economic value

We estimated the SCC associated with NWR-specific NECB. This simple calculation is accomplished by converting NECB to its $CO_2$ equivalent and then applying the SCC per ton of $CO_2$ to derive the monetary value associated with additional or avoided $CO_2$ emissions in a given year. We assumed no leakage of $CO_2$ emissions as a result of human disturbance because NWRS generally do not allow these activities on these public lands. The relative benefits of avoided emissions vary based on when they are achieved due to the persistence of $CO_2$ emissions in the atmosphere, time until climate change effects are expected, and discounting of future benefits. Discounting of future costs and benefits is a standard practice in economics, which accounts for individuals' preference to avoid current damages over future damages (i.e., the value placed on a benefit today is higher than the value of a benefit enjoyed a year from now). Discounting allows for the comparison of benefits and costs that occur in different time periods by normalizing their values (i.e., placing all values in a standard value term) [10]. Since the NECB estimates are annual estimates, we use SCC values for 2018. Actual NECB and

associated SCC values would vary across time due to normal fluctuations in NWR carbon sequestration and with changes in the SCC valuation.

Discount rates have a considerable influence on the results of valuation for long time horizon impacts such as those associated with climate change. There is a lack of consensus on the discount rate, therefore the Interagency Working Group [30] recommends considering a range of four standard discount rates: 5%, 3% and 2.5% discount rates associated with average damage (determined by impact models) and a 3% discount rate with damage at the 95th percentile. This fourth estimate represents the least likely, but highest impact climate change scenario (see [30]) for more information on damages modeling and discount rates). We consider all four discount rates/scenarios in our analysis. The initial SCC estimates were provided in 2007 USD, so we adjusted for inflation to 2018 USD using the Consumer Price Index. The mid-range 3% discount rate for avoiding an additional ton of $CO_2$ in 2018 was $48.40 with 2.5% and 5% rates at $73.0 and $15.0, respectively for average climate change scenario damages. Fig 3 illustrates the increasing value associated with C sequestration for each of the four discount rate scenarios. The values are represented as an annual net present value and the increase is largely due to the increasing climate change effects as the models predict the atmospheric concentration increases through time. The mid-range 3% discount rate value nearly doubles between 2018 and 2050.

## 3. Results

### 3.1. Effects of refuge protection on carbon resources

Area, mean carbon stock, and mean NECB are summarized in Table 1 with the final results of the two-way ANOVA analysis for the two carbon variables (mean stock and NECB) listed in

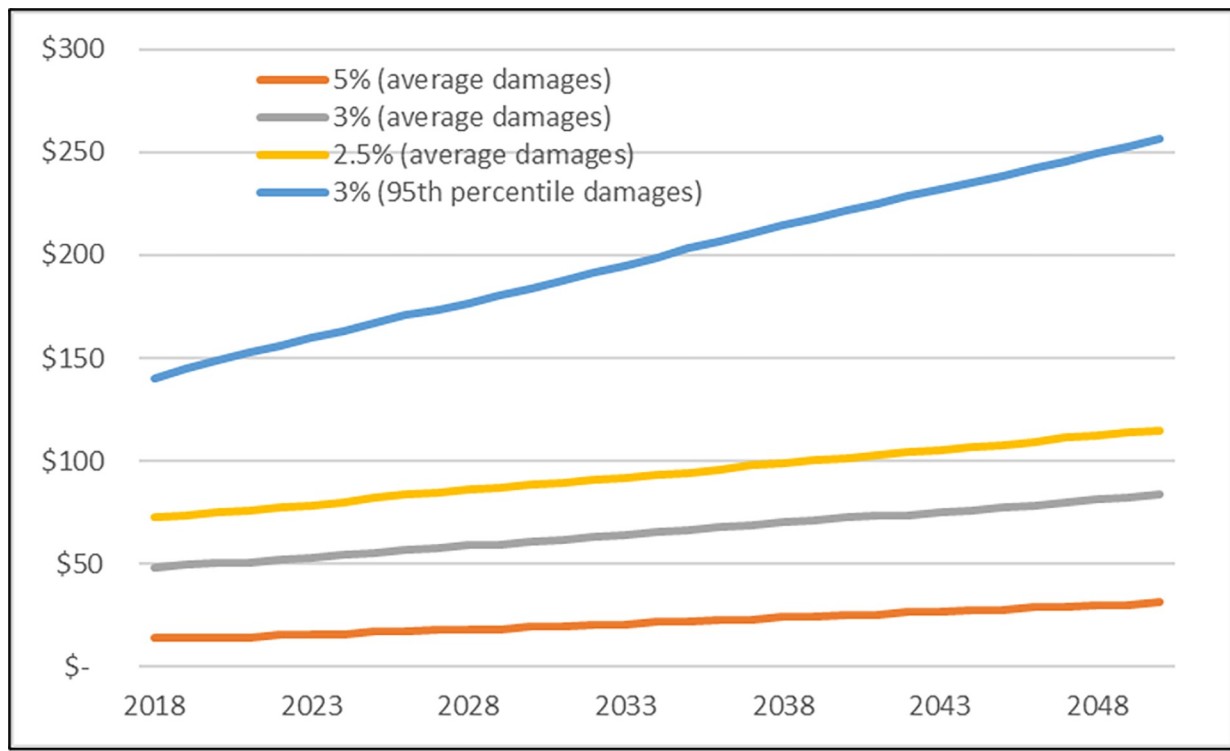

**Fig 3. Social cost of carbon per ton of $CO_2$ equation for 2018–2050.** Values represent net present value of damages avoided from one ton of carbon dioxide at the 2.5, 3, and 5 percent discount rates for average climate change scenario damages and at the 3% discount rate for the 95th percentile climate change scenario damages. Estimates have been escalated to 2018 USD using the Bureau of Labor Statistics Consumer Price Index. Original values from IWG (2016).

**Table 2. Final results of two-way Analysis of Variance (ANOVA) for two variables: a) mean carbon stock, and b) log mean net ecosystem carbon balance (NECB).**

| Main effects | df | Chi-square or F | P | Significance |
|---|---|---|---|---|
| a) Log of refuge area | 76 | 1019.1 | <0.0001 | *** |
| Total area | 1 | 146.6 | <0.0001 | *** |
| Region | 8 | 4.4 | <0.0001 | *** |
| Total area * region | 7 | 43.6 | <0.0001 | *** |
| Inside-outside refuge | 1 | 274.5 | <0.0001 | *** |
| Region * inside-outside_refuge | 8 | 3.8 | 0.0002 | *** |
| Dominant habitat nested within region | 21 | 7.5 | <0.0001 | *** |
| Dominant_habitat inside-outside_refuge | 22 | 5.2 | <0.0001 | *** |
| nested in region | | | | |
| Lyears | 1 | 4.1 | 0.0446 | * |
| Lyears * region | 7 | 4.0 | 0.0003 | *** |
| b) Log mean carbon stock | 78 | 1025.4 | <0.0001 | *** |
| Region | 8 | 7.5 | <0.0001 | *** |
| Total_area | 1 | 8.3 | 0.0040 | ** |
| Total area * region | 7 | 5.6 | <0.0001 | *** |
| Inside-outside refuge | 1 | <0.1 | 0.9064 | n.s. |
| Region * inside-outside refuge | 8 | 0.4 | 0.9342 | n.s. |
| Dominant_habitat nested in region | 21 | 15.0 | <0.0001 | *** |
| Dominant_habitat * inside-outside refuge nested in region | 22 | 0.4 | 0.9961 | n.s. |
| Lyears | 1 | 0.2 | 0.6922 | n.s. |
| Lyear * region | 7 | 6.4 | <0.0001 | *** |
| Latitude | 1 | 8.3 | 0.0041 | ** |
| Longitude | 1 | 35.0 | <0.0001 | *** |
| c) Mean NECB | 59 | 918.3 | <0.0001 | *** |
| Region | 8 | 9.6 | <0.0001 | *** |
| Inside-outside refuge | 1 | 1.3 | 0.2484 | n.s. |
| Region * inside-outside refuge | 8 | 0.2 | 0.9821 | n.s. |
| Dominant habitat nested in region | 22 | 8.8 | <0.0001 | *** |
| Dominant habitat nested in region * inside-outside refuge (region) | 20 | 0.4 | 0.9909 | n.s. |

The models tested included a) model area_refuge = total_area|region region|inside-outside_refuge Dominant_habitat(region)| inside-outside refuge lyears|region, b) model mean carbon stock_refuge = region| inside-outside refuge Dominant_habitat(region)| inside-outside refuge, and c) model lmean_carbon_stock_refuge = latitude longitude total_area|region region| inside-outside_refuge Dominant_habitat(region)| inside-outside_refuge lyears|region. Model fits for the three models are: Chi-square = 1019.1, 1025.4 and 918.3, respectively, p < 0.0001. ANOVA tests were based on Proc Mixed in SAS (2012) with main effects based on Type III Sums of Squares (SAS, 2012). Significance differences are denoted by "*", "**", and "***" at p < 0.05, p < 0.01 and p < 0.001, respectively. Significant interactions of main effects are shown for mean carbon stock and flux in Figs 4 and 5, respectively (p < 0.0001).

Table 2. The test of whether there was a difference between habitats within refuges and the lands immediately outside of the refuges was significant. The effect of protection status was influenced by the size of the area protected but was not related to latitude and longitude (p > 0.05). While the mean values of carbon stock and NECB were higher inside of the refuges (10,957 $gCm^{-2}$ and 102 $gCm^{-2}yr^{-1}$, respectively) than the lands outside (9,993 $gCm^{-2}$ and 85 $gCm^{-2}yr^{-1}$, respectively; p < 0.0001) (S1 Table), the values of carbon stock and NECB inside and outside of the 364 refuges did not differ (p > 0.05; Table 2). In our analysis, we also considered the role of area sizes as a potential covariate in the variability found in carbon stock and NECB. The analysis showed that the mean carbon stock differed by area sizes by regions, larger sizes offered more protection to carbon stock (Table 2).

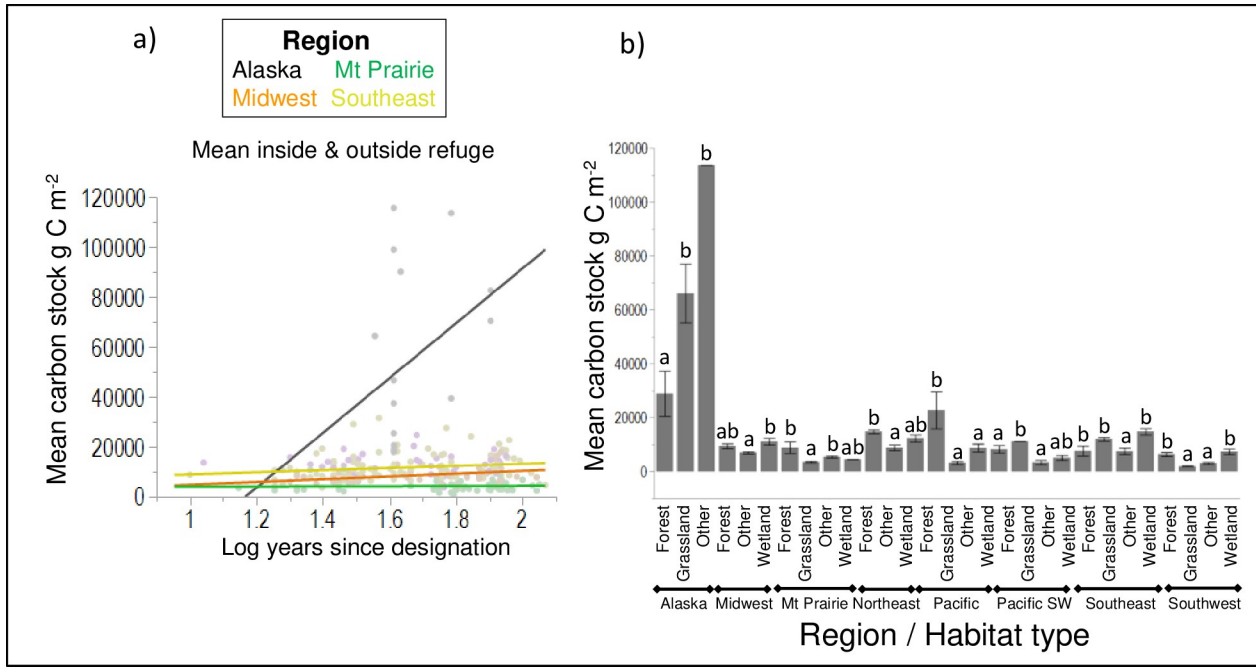

**Fig 4.** Regional variation in mean carbon stock (gCm⁻²) using combined means inside and outside of the refuge by a) time since refuge designation and b) dominant habitat by regions with significant differences of multiple mean comparisons after Bonferroni correction indicated using different letters. Mean carbon stock increased over time since refuge designation in regions including Alaska, Midwest, Mt Prairie, and Southeast (p < 0.05, Table 3). Regions that did not change over time (not shown in the figure) included Hawaii, Northeast, Pacific Southwest, Pacific, and Southwest (p > 0.05). Hawaii did not vary by dominant habitat type (p > 0.05) and thus not listed, while the other regions did vary by dominant habitat type. Fit of the overall model was significant (Chi-square = 1005.9, p < 0.0001; Table 2).

### 3.2. Carbon stock by service region over time

The relationship of carbon stock to the time of refuge designation differed significantly by region (Table 2; p < 0.0001) and was also influenced by latitude and longitude (p = 0.0041 and p < 0.0001, respectively; Table 2). The Alaska, Midwest, Mt. Prairie, and Southeast regions had had higher carbon stock in refuges that were designated earlier than those designated more recently (Fig 4). In other regions carbon resources were stable (did not differ) within and outside of the refuge designations (p > 0.05). As can be expected, carbon stock had a distinctive pattern with habitat types within regions (p < 0.0001; Table 2). For example, most NWRS carbon resources are stored in refuges in the Alaska Region, while refuges in the Southeast Region are most productive with the highest mean NECB (Table 1). Grasslands hold more carbon stock than other habitat conditions within the NWRS due to the fact that it has more land area than any other habitat (Table 1). Such patterns explain different and unique carbon resources managed by FWS as public lands and the information could be useful for management prioritizations.

### 3.3. Carbon balance by service region and habitat type over time

Mean NECB, which summarizes the balance of major carbon fluxes in ecosystems, differed by region depending on habitat type (p < 0.0001), but was not related to geographic location (latitude and longitude) (p>0.05; Table 2). Not surprisingly, refuges dominated by forest habitat generally had higher mean NECB (stronger carbon sink) than those dominated by grassland, wetland, and other habitat types (Fig 5). For example, refuges in the Midwest Region dominated by forest had higher NECB than wetland. Similarly, refuges dominated by forest habitat

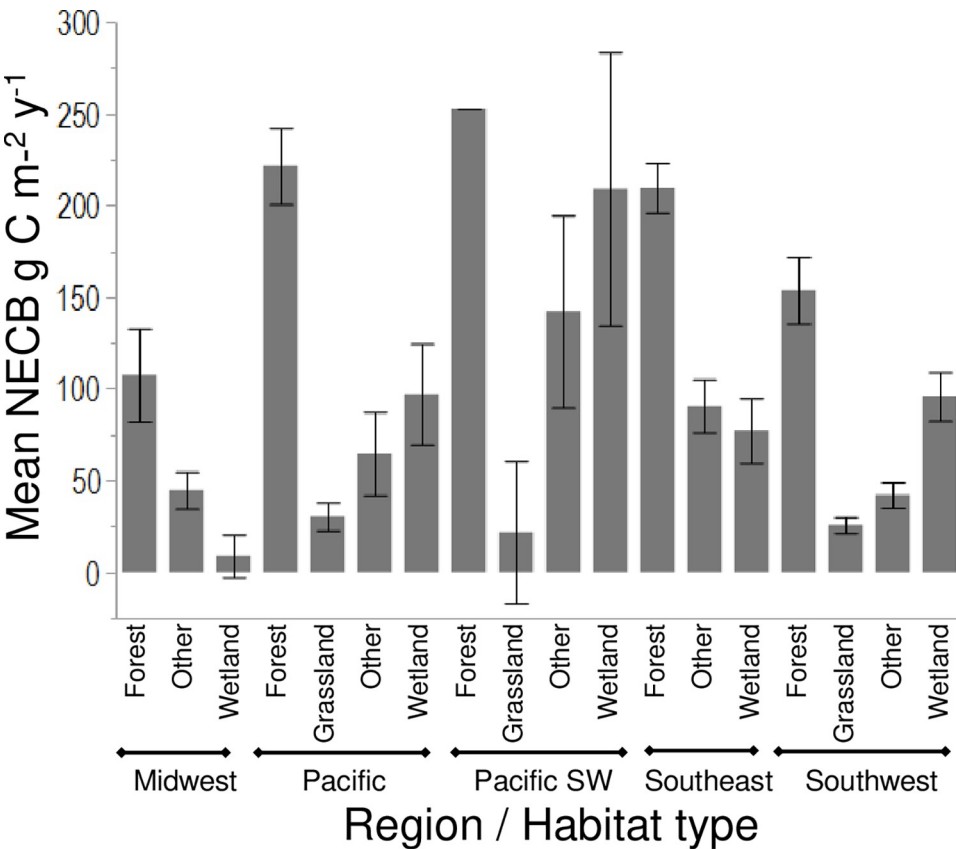

**Fig 5. Comparison of net ecosystem carbon balance (NECB) (gCm$^{-2}$yr$^{-1}$) by region and habitat type (p $<$ 0.0015) using combined means of habitats inside and outside of refuge (p $>$ 0.05).** Significant differences of multiple mean comparisons after Bonferroni correction are indicated using different letters. Regional habitats not shown were not significantly related to the ANOVA model and included Alaska, Hawaii and Mountain Prairie (p $>$ 0.05). The fit of the overall model was significant (Chi-square = 1023.9, p $<$ 0.0001; Table 2).

had higher NECB than other habitat types in other regions such as Pacific, Pacific Southwest, Southeast, and Southwest (Fig 5). However, unlike its effects on carbon stock, the variable of years since refuge designation by regions did not test significantly for NECB, suggesting that the overall balance of carbon fluxes in the refuges could be influenced by other, more complex factors such as natural disturbances, or aging of forest habitats.

### 3.4. Economic value

The SCC varied across regions and ecosystem types as shown in Table 3. Values represent the net present value of damages avoided from $CO_2$ emissions for the 3% discount rate (results for all discount rates are available in S1 Table). Positive values indicate net C sequestration and associated benefits while negative values indicate net C emissions and associated costs. All of the variation is due to the underlying NECB as the SCC value is the same across regions (as a single global estimate of damages). The entire NWRS provides an annual net benefit of $976 million in terms of avoided $CO_2$ emissions at the 3% discount rate. The SCC ranges from $292 million (at the 5% discount rate) to as much as $2.8 billion for the 3% discount rate at the 95th percentile (i.e., in the less likely, but far more costly damages estimate). In terms of habitats, forests provide the highest value (US $360 million per year), and wetlands the smallest value (US $65 million per year).

**Table 3. Total and mean (per square km) social cost of carbon by region and ecosystem type.**

| Region | Forest | | Grassland | | Wetland | | Other | | Regional summary | |
|---|---|---|---|---|---|---|---|---|---|---|
| | Total | Mean | Total | Mean | Total | Mean | Total | Mean | Total | Mean |
| Pacific | 10 | 42.7 | 13 | 2.9 | 4 | 21.6 | 8 | 7.0 | 35 | 5.8 |
| Southwest | 27 | 31.2 | 29 | 3.3 | 27 | 17.7 | 6 | 8.3 | 89 | 7.4 |
| Midwest | 27 | 16.9 | | | -2 | -2.8 | 12 | 9.1 | 37 | 10.5 |
| Southeast | 370 | 43.5 | | | 14 | 6.5 | 34 | 22.4 | 418 | 34.2 |
| Northeast | 37 | 24.5 | | | 5 | 15.0 | 4 | 24.8 | 46 | 22.8 |
| Mountain Prairie | 0 | -2.6 | -10 | -1.5 | 0 | 5.8 | -3 | -2.4 | -13 | -1.6 |
| Alaska | -112 | -1.3 | 386 | 1.6 | | | 11 | 6.8 | 285 | 0.9 |
| Pacific SW | 1 | 47.8 | 45 | 6.0 | 16 | 34.0 | 16 | 17.5 | 78 | 8.8 |
| Hawaii | 0 | 0.1 | 0 | 45.9 | | | 0 | 61.1 | 0 | 2.1 |
| Habitat summary | 360 | 3.6 | 463 | 1.7 | 65 | 12.0 | 87 | 10.1 | 976 | 2.5 |

Total values are in 2018 million USD, mean values are in 2018 thousand USD. Values represent the net present value of damages avoided from carbon dioxide using a mid-range 3% discount rate for average climate change scenario damages. Positive values indicate net C sequestration and associated benefits while negative values indicate net C emissions and associated costs. Blank table cell indicates that the habitat type does not exist in the region from the data used.

On a per unit basis, there is heterogeneity in the value of the SCC per square kilometer by region and ecosystem type. This variation is due to the differing rates of productivity across regions and ecosystems (i.e., a forest in Hawaii may be far more productive leading to greater $CO_2$ sequestration than one in the Northeast). Forest SCC values in the Pacific Region are US $42,696 per square kilometer while in the Mountain Prairie Region the value is US -$2.6. Conversely, grasslands in the Pacific provide US $2.9 per square kilometer while in the Mountain Prairie Region the SCC is US -$1.5 per square kilometer. It should be noted that these values are only as precise as the land cover data used in the analysis; actual NECB by habitat is likely to vary within regions.

For individual NWRS, the range in SCC is from US $130 million per year in emissions costs from the Yukon Flats NWR to $149.9 million per year in avoided emissions benefits from the Alaska Peninsula NWR (for the average damages 3% discount rate). Results for the individual NWRS at the 2.5, 3, and 5% discount rates for average climate change scenario damages and at the 3% discount rate for the 95th percentile climate change scenario damages are available in S1 Table. The Great Dismal Swamp NWR in Virginia and North Carolina provides an estimated $11 million in annual C emission reduction benefits; these values are comparable to annual values estimated by [9], which ranged from -$5 to $24 million in annual benefits. That study used additional field and literature information and was based on a micro-scale analysis of C sequestration in the refuge.

## 4. Discussion

### 4.1. Carbon storage in refuges

Public lands such as NWRS protect forests, grasslands and wetlands managed for public benefit within the federal system provide valuable ecosystem services related to wildlife habitat, clean water, storm barriers and public recreational opportunities [31]. In addition to these benefits, our analysis shows that these lands provide additional public benefit of sequestering carbon to mitigate atmospheric $CO_2$ increases (Table 3). A substantial amount of carbon stock (16.6 PgC, Table 1: overall mean stock multiplying with total area) is stored within the NWRS in grassland, forest, wetland and other ecosystem types resulting in positive economic benefits (for a total of at least $976 million per year) for the public related to SCC (Table 3). The similar

benefit of public land has been documented for national parks within the U.S. National Park system with an annual benefit of \$582.5 million [8]. Quantifying the current SCC benefits of the refuge system and the relative values in ecosystem types and regions supports decisions on conservation and potentially, acquisition of additional land as part of a portfolio of ecosystem services.

## 4.2. Management and carbon in refuges

Conceptually, the majority of habitat types on NWRS lands would accumulate more carbon in older refuges with mature vegetation (i.e., refuges designated many decades earlier). This idea is supported by the result of this study showing that the age of refuge has an overall positive effect on the amount of carbon stored in habitats i.e. older refuges generally store more carbon than younger ones. Regionally, the age-carbon stock relationship is evident in several FWS regions (such as Alaska, Mt. Prairie, Midwest, and Southeast, Fig 4), but has a more stable pattern in other regions. Age effect on ecosystem carbon fluxes (NECB) is not significant. This pattern is understandable given the effects of NECB saturation over time among different ecosystems [32, 33]. Natural disturbances such as wildfire and drought have a large impact on NECB variations. Episodic disturbance events in the western United States and Alaska, such as major droughts and wildfires, have had strong effects on a downward trend in the strength of NECB [34, 35]. Wildfires in eastern peatlands have also resulted in a significant reduction of carbon stocks (e.g., Great Dismal Swamp NWR, Okefenokee NWR, Pocosin Lakes NWR).

However, a separate question we asked is: are the age effects on the two carbon variables the result of protection status in refuges? In other words, is there a carbon difference between habitats within the refuges and lands outside? On this, our results are mixed. The average carbon stock and average NECB are higher (10,957 $gCm^{-2}$ and 102 $gCm^{-2}yr^{-1}$, respectively) in refuges than the lands immediately outside of refuges (9,993 $gCm^{-2}$ and 85 $gCm^{-2}yr^{-1}$, respectively, S1 Table). Therefore, on average, NWS refuges store 969 more grams of carbon per square meter than the lands outside, and their carbon sink is greater inside than outside by 15 grams per square meter per year, even though we did not find a statistical difference. There could be several reasons contributing to the test results, including the possibility of conditions immediately outside of refuge lands still retaining similarities of the refuge in terms of vegetation biomass and soil attributes. We did not further analyze possibilities of whether there is a general pattern of significant land-use change immediate outside of refuge lands. Our findings broadly suggest that carbon balance may be vulnerable to change in the future with changes in environment and climate; however, opportunities exist at individual refuges and regions where it is possible to increase carbon resources under protection or management (see Table 4).

An important lesson from this study is that publicly managed lands hold large amounts of carbon resources, which provide mitigation benefits to society. Thus, efforts to enhance carbon resources through protection and management may be useful as illustrated in Table 4. In newly designated refuges, replanting establishes carbon stock more quickly, particularly in grassland and forest habitat. Old agricultural or pastoral fields can be revegetated with native vegetation with increased long-term ecosystem services and carbon sequestration accrued over the decades following public land designation. In Neal Smith NWR, investigators documented increases in soil organic carbon in newly established tallgrass prairie on previously cultivated land (e.g., [36]). Restoration projects in refuges have many other benefits including storing floodwater, reconnecting small parcels of wildlife habitat to establish corridors, and protecting land from development. The Service has been instrumental in such restoration efforts in collaboration with several partners including The Conservation Fund (TCF; *Go Zero*® Program), the Trust for Public Land (TPL), and cooperating farmers (USFWS, 2017; Table 4). Note that

**Table 4. Replanting programs on U.S. Fish and Wildlife Refuges by region and location, projected carbon sequestration in the next 50 and 100 years, project standard set by SCS global services (SCS) (Gold or other level) and year of designation by the Climate, Community, and Biodiversity Alliance (CCBA) [48].**

| FWS refuge | Location | Vegetation replanted | Acres replanted | Restoration partners | Carbon sequestered 50 y (MT) | Carbon sequestered 100 y (mT) | CCBA standard | SCS verification | Ecosystem services & notes |
|---|---|---|---|---|---|---|---|---|---|
| Upper Ouachita | NW LA, Region 4 | Bottomland hardwood | 2,600 | TCF *Go Zero* | 675,666 | 851,793 | 8/2011; Gold | 7/2016 | Reduce flooding in communities, e.g., 2010. |
| Lake Ophelia & Grand Cote | Central LA, Region 4 | Bottomland hardwood & cypress | 814 | TCF *Go Zero* | 211,048 | 266,580 | 12/2010; Gold | 7/2015 | Reconnected forest fragments for black bear. |
| Red River | NW LA, Region 4 | Bottomland hardwood | 1,170 | TCF *Go Zero* | 306,461 | 387,419 | 5/2009, Gold | 4/2014 | Biodiversity protection from development. |
| Tensas River | N LA, Region 4 | Bottomland hardwood | 8,000 | TPL | 484,841 | 612,922 | | * | Carbonfund.org prepared a Project Design Document (PDD) for CCBA Standards in 2008–2009. Reconnected forest fragments for black bear. |
| Mingo | SE MO, Region 4 | Bottomland hardwood & cypress | 367 | - | 95,153 | 120,290 | 5/2010, Gold | 5/2010 | |
| South Texas Refuge Complex | S TX, Region 2 | Tamaulipan thorn scrub | 2,000 | TCF, American Forest Global ReLeaf & cooperating farmers | 20,500[a] | 28,600[b] | - | - | TCF and American Forests' Global ReLeaf |
| Marais de Cygnes | KS, Region 6 | riparian hardwood forests & tallgrass prairie | 776 | TCF & Environmental Synergy Inc. | 192,185 | 357,871 | 7/2009, Gold | 5/2014 | River protection. |
| Great Dismal Swamp | | | | | | | | | Reduce risk of catastrophic fire. |

Partners include The Conservation Fund (TCF) *Go Zero*® Program, Trust for Public Land (TPL), and cooperating farmers (individuals). CCBA is the Climate, Community, and Biodiversity Alliance and SCS is the Scientific Certification Systems Global Services. Both companies do third-party certification and verification of environmental and business capability and outcome.

*CCBA standards were verified by the Rainforest Alliance (~2008–2009); Projected carbon sequestration in [a]20 years; [b]40 years.

when a refuge acquires agricultural lands, the soil organic carbon accumulation rate may decrease some years after a shift from cropland until natural vegetation redevelops [37].

Beyond restoration via replanting, carbon management could be enhanced further through activities designed to maximize carbon stock held in vegetation, detritus, and soil [38]. In wetland ecosystems examined in this study, soil harbors the largest long-term carbon pool. Various management techniques might increase levels of carbon storage, but among the best are those that maximize production levels of native vegetation [20, 38]. In wetlands such as the Great Dismal Swamp (Table 4), re-wetting peat wetlands after the removal of drainage ditches can lower the risk of catastrophic fires (Fig 2), increase native species dominance, and increase carbon sequestration [9, 39]. Additional public health benefits are accrued through peat rewetting to reduce wildfire frequency and intensity [40]; previous studies have documented public health impacts of peat carbon emissions related to a Pocosin Lakes NWR wildfire (e.g., [41]). Studies have also shown that impoundment reduces the level of production in wetlands [42, 43]. Studies such as [44] determined that natural tidal freshwater marshes had significantly higher carbon storage and vertical accretion rates than impounded and seasonally drained

marshes. The results strongly suggest that "the long drainage period in moist soil management limits carbon storage over time."

The restoration of flood pulsing in riverine wetlands [45] or tidal pulsing in salt marshes via the removal of levees and dikes [38] can be key to restoring vegetation health, production, and ultimately long-term carbon stocks in the soil. In the Nisqually River Delta in Puget Sound, Washington, Service land managers in Billy Frank Jr. Nisqually NWR removed an extensive dike in 2009 and restored tidal flow to 308 ha of the refuge, while in the same timeframe the Nisqually Tribe restored 57 ha in the delta. A USGS collaborative investigation determined that there was a net of 105.4 ha of emergent marsh wetland within the Nisqually River Delta between 1957 and 2015, largely as a result of restoration efforts that occurred in several phases through 2009 [46]. In addition, the study also found that a restored marsh can quickly begin to accrete sediment and store carbon, even with sparse plant colonization; however, the accumulated carbon in the sparsely vegetated site may be more vulnerable to erosive loss [47]. The study determined that "restored and historic marshes can have similar carbon accumulation rates even with divergent marsh formation processes. This study provides empirical evidence that the management of wetland habitats via the removal of structures that impede flood or tidal flow may ultimately contribute to better carbon conservation. Our research suggests that in view of the potential of public land to sequester carbon in the long term, the discussion of the best management practices of natural lands is warranted as part of the national discussion of carbon mitigation.

The annual benefits of these public lands in carbon sequestration are worth over $600 million with the potential to increase the avoided SCC contribution via management actions. See [9] for a comparison of carbon sequestration values under different management scenarios. Thus, public lands may provide public benefits beyond widely appreciated environmental amenities, particularly by offsetting emissions of other sectors. Considering their societal value, NWRS may elevate the importance of carbon management and support innovative funding mechanisms to conserve and restore public lands.

## 5. Conclusion

Public lands offer many ecosystem services to society including carbon sequestration to mitigate the effects of climate change. This study, using the United States National Wildlife Refuges as a test case and based on existing data sources, revealed that the age of lands designated as public lands (thus subject to policies of conservation and protection) has an effect on increased carbon stock compared to lands outside of the refuge boundaries. The effect is variable by regions and habitat types. The rate of carbon sequestration by the NWRS is worth $976 million annually calculated as the social cost of carbon with a 3% discount rate. We also highlighted examples of carbon management occurring in individual NWR lands.

## Supporting information

**S1 Table. Mean carbon stock and mean net ecosystem carbon balance in and outside of the national wildlife refuges studied, total carbon dioxide sequestered, and associated social cost of carbon for National Wildlife Refuges.** Valuation for social cost of carbon based on the Interagency Working Group on social cost of carbon values [30] escalated to 2018 USD using the Bureau of Labor Statistics Consumer Price Index and the net ecosystem carbon balance. Values represent net present value of damages avoided at the 5, 3, and 2.5 percent discount rates for average climate change scenario damages and at the 3% discount rate for the 95th percentile climate change scenario damages.
(DOCX)

## Acknowledgments

We thank anonymous reviewers for advice on earlier drafts of this manuscript. The views and conclusions in this article represent those of the authors from U.S. Fish and Wildlife Service and the U.S. Geological Survey, while at the same time adhering fully to the standards of the Fundamental Science Practices of the U.S. Geological Survey. Any use of trade, firm, or product names is for descriptive purposes only and does not imply endorsement by the U.S. Government.

## Author Contributions

**Conceptualization:** Zhiliang Zhu, Beth Middleton, Emily Pindilli, Kurt Johnson.

**Data curation:** Kurt Johnson.

**Investigation:** Beth Middleton, Emily Pindilli.

**Methodology:** Zhiliang Zhu, Beth Middleton, Emily Pindilli, Darren Johnson.

**Supervision:** Zhiliang Zhu.

**Validation:** Beth Middleton.

**Writing – original draft:** Zhiliang Zhu, Beth Middleton, Emily Pindilli, Kurt Johnson, Scott Covington.

**Writing – review & editing:** Zhiliang Zhu, Beth Middleton.

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
