## [Decision Letter · Decision Letter 0]

22 Jun 2021

PONE-D-21-13289

Conservation of carbon resources and values on public lands: A case study from the National Wildlife Refuge System

PLOS ONE

Dear Dr. Zhu,

Thank you for submitting your manuscript to PLOS ONE. After careful consideration, we feel that it has merit but does not fully meet PLOS ONE’s publication criteria as it currently stands. Therefore, we invite you to submit a revised version of the manuscript that addresses the points raised during the review process.

ACADEMIC EDITOR: Please revise taking in to account the reviewers comments. Please improve the English usage in the manuscript.

We look forward to receiving your revised manuscript.

Kind regards,

Arun Jyoti Nath

Academic Editor

PLOS ONE

Journal Requirements:

3.We note that Figure #1 in your submission contain map images which may be copyrighted. All PLOS content is published under the Creative Commons Attribution License (CC BY 4.0), which means that the manuscript, images, and Supporting Information files will be freely available online, and any third party is permitted to access, download, copy, distribute, and use these materials in any way, even commercially, with proper attribution. For these reasons, we cannot publish previously copyrighted maps or satellite images created using proprietary data, such as Google software (Google Maps, Street View, and Earth). For more information, see our copyright guidelines: http://journals.plos.org/plosone/s/licenses-and-copyright.

a. You may seek permission from the original copyright holder of Figure #1 to publish the content specifically under the CC BY 4.0 license.  

4. We note that Figure #2 in your submission contain copyrighted images. All PLOS content is published under the Creative Commons Attribution License (CC BY 4.0), which means that the manuscript, images, and Supporting Information files will be freely available online, and any third party is permitted to access, download, copy, distribute, and use these materials in any way, even commercially, with proper attribution. For more information, see our copyright guidelines: http://journals.plos.org/plosone/s/licenses-and-copyright.

a. You may seek permission from the original copyright holder of Figure #2 to publish the content specifically under the CC BY 4.0 license. 

Reviewers' comments:

Reviewer's Responses to Questions

**Comments to the Author**

1. Is the manuscript technically sound, and do the data support the conclusions?

Reviewer #1: Yes

Reviewer #2: Yes

2. Has the statistical analysis been performed appropriately and rigorously? 

Reviewer #1: Yes

Reviewer #2: No

3. Have the authors made all data underlying the findings in their manuscript fully available?

Reviewer #1: Yes

Reviewer #2: Yes

4. Is the manuscript presented in an intelligible fashion and written in standard English?

Reviewer #1: Yes

Reviewer #2: Yes

5. Review Comments to the Author

Reviewer #1: Introduction

• Introduction is logically prepared and well written

• Highlight the significance of the research in the context of changing climate

Materials and Methods

• Data on carbon stock and net ecosystem carbon balance were found as secondary source, I doubt about that quality of data and there analysis method are not mentioned in the section. Please include all the information and provide the data source as supplementary materials

Reviewer #2: Pl see attachments. The paper addresses important aspects and needs to be modified further. The Introduction (justification and uses of the study); method (testing of ANOVA assumptions); result modification and complete section of conclusion should be added, please.

6. PLOS authors have the option to publish the peer review history of their article (what does this mean?). If published, this will include your full peer review and any attached files.

Reviewer #1: No

Reviewer #2: **Yes: **Rajiv Pandey

---

## [Author Response · Author response to Decision Letter 0]

21 Oct 2021

1. The language needs to be modified.

Response: This comment and the next one below recommends that we edit the manuscript for language purposes. We agree. We conducted a complete read-through and provided updates (shown with track changes). In addition, the manuscript is a US government information product and was already subjected to 1) an internal technical review, and 2) a government approval process before our submission to the journal. 

2. The editing for MS is desired. I suggest to see the MS for meticulous way. (Comma, mixing or joining or words etc.) 

Response: please see above.

3. The abstract should explain the method. 

Response: Agreed. We have updated the abstract, please see track changes throughout the abstract that provided details of the methods.

4. The keywords should not include the words of the Title of MS.

Response: We have modified the keywords, which now do not include words already used in the title.

5. Please define public lands focusing to the study.

Response: The term is defined in text (first sentence and reference in Introduction). But it was not defined in the abstract. We have added the definition in abstract (first sentence). 

6. Please include one para at the end of Introduction about the uses of the study.

Response: We have added the following paragraph: Public lands have traditionally been managed to support a variety of uses including conservation and recreation. Increasingly they are also being looked upon to provide benefits through climate change mitigation via carbon sequestration and reduction of greenhouse gas emissions. The purpose of this study was to add useful information to the new management objective.

7. The study area along with the detailed NWRS should be included as supplementary material. This may include longitude and latitude, major species, areas, year of establishment, management regime. etc.

Response: We have updated the supplementary material table and provided latitudes and longitudes of the centers of the NWR, along with major land cover, areas, and year of establishment. We also added latitudes and longitudes to the statistical analysis, which improved the power of the statistical analysis by reducing unexplained variation of the model.

8. Please provide details of two factors for ANOVA along with main effect and interaction. 

Response: We ran a nested design with three factors: region, areas inside or outside of the refuge, and dominant habitat, plus any significant covariables (with latitudes and longitudes added) based on a likelihood ratio test. The covariables were tested both in and out of the selected model. The final model is included in Table 2 in the manuscript.

9. Have you tested the assumptions of ANOVA specially linearity and homogeneity. Please elaborate.

Response: Linearity was tested with the chi-square Likelihood ratio test between the full and null model. Homogeneity was tested looking at the residuals of the model. The residuals had a random pattern. 

10. Please elaborate the changes per unit area. Please provide the details of ANOVA.

Response: All of our results are given either as per unit area (such as Fig. 4 and 5, and Table 2), or can be calculated as such (such as Table 1 and 3, and as supplementary material where the total area is given; division will give per unit area results). Details of ANOVA including nonsignificant results are provided in Table 2.

11. Please see with longitude and latitude the changes are significant.

Response: Latitudes and longitudes were only significant with carbon stock, but not with the two other factors (area and NECB), see Table 2.

12. The discounting approach is straight however forests are dynamic. Please see if you can estimate the deforestation and degradation aspect of the forest. This can be achieved by analyzing the forest cover in successive years. This will add the dynamic components of the change.

Response: This study included all major habitat types (land cover types) such as forests, wetlands, grasslands, agricultural lands. In fact, forest land cover is a small part of the United States National Wildlife Refuge System. The existing data we used already included forest cover change as well as dynamic changes in other habitat types from the original study (see the reference: Zhu and Stackpoole 2011).

13. How the social cost for future has been estimated. Business usual scenario may not exist in future. Please see if population changes or management regime changes. (what about fire as mentioned in the MS)

Response: Social cost of carbon is defined as an estimate of the net present value of avoided economic and societal damages associated with increase (or decrease) in CO2 emissions for a given year. The value is associated with discount rates which are used to project for future years. The discount rates are broad enough to include a range of scenarios including business as usual scenario (3% discount rate is the middle of the road scenario). Population change or management regime change (including wildfire management) affect carbon sequestration, which affect the estimate of social cost of carbon.

14. Can you discuss about the management of these lands, as the management may also influence the carbon capturing.

Response: Management of the public lands is the primary focus of the study. Management objectives are described throughout the manuscript. For example, in the 2nd paragraph of Introduction, it states: National Wildlife Refuges (NWRS) are managed by the U.S. Fish and Wildlife Service to protect habitat following laws such as the National Wildlife Refuge System Improvement Act, Migratory Bird Treaty Act and the Endangered Species Act.

15. The ecosystem services (additional) may be discussed in terms of added economic gains, if possible.

Response: We have discussed ecosystem services throughout the manuscript in terms of benefits provided to users of public lands such as recreation and climate mitigation. However, “economic gains” are not included in the discussion because, by law, public lands in the United States are not managed for economic gains. 

16. Please add a section for conclusion.

Response: We have added a Conclusion section in addition to Discussion. 

17. Please add review of literature on the aspect in Introduction.

Response: Can you be more specific about the topic of any missing information? There is a thorough review of literature on the topic of this manuscript (public land management and carbon sequestration) in the Introduction.

---

## [Editor Report · Decision Letter 1]

20 Dec 2021

Conservation of carbon resources and values on public lands: A case study from the National Wildlife Refuge System

PONE-D-21-13289R1

Dear Dr. Zhu,

We’re pleased to inform you that your manuscript has been judged scientifically suitable for publication and will be formally accepted for publication once it meets all outstanding technical requirements.

Kind regards,

Arun Jyoti Nath

Academic Editor

PLOS ONE
---

## [Editor Report · Acceptance letter]

4 Jan 2022

PONE-D-21-13289R1 

Conservation of carbon resources and values on public lands: A case study from the National Wildlife Refuge System 

Dear Dr. Zhu:

I'm pleased to inform you that your manuscript has been deemed suitable for publication in PLOS ONE. Congratulations! Your manuscript is now with our production department. 

Kind regards, 

on behalf of

Dr. Arun Jyoti Nath 

Academic Editor

PLOS ONE